# Public Health Needs the Public Trust: A Pandemic Retrospective

Matthew T. J. Halma [1]  and Joshua Guetzkow [2,*]

1   EbMC Squared CIC, Bath BA2 4BL, UK
2   Institute of Criminology, Department of Sociology & Anthropology, The Hebrew University of Jerusalem, Jerusalem 91905, Israel
*   Correspondence: joshua.guetzkow@mail.huji.ac.il

**Abstract:** The COVID crisis of the past three years has greatly impacted stakeholder relationships between scientists, health providers, policy makers, pharmaceutical industry employees, and the public. Lockdowns and restrictions of civil liberties strained an already fraught relationship between the public and policy makers, with scientists also seen as complicit in providing the justification for the abrogation of civil liberties. This was compounded by the suppression of open debate over contentious topics of public interest and a violation of core bioethical principles embodied in the Nuremberg Code. Overall, the policies chosen during the pandemic have had a corrosive impact on public trust, which is observable in surveys and consumer behaviour. While a loss of trust is difficult to remedy, the antidotes are accountability and transparency. This narrative review presents an overview of key issues that have motivated public distrust during the pandemic and ends with suggested remedies. Scientific norms and accountability must be restored in order to rebuild the vital relationship between scientists and the public they serve.

**Keywords:** public health; public trust; science communication; pedagogy; citizen science; stakeholders; informed consent; uncertainty communication



## 1. Introduction: A Loss of Trust

The response to the emergence of SARS-CoV-2 has had a significant impact on the relationship between the biomedical community and the public. Public perceptions impact the decision to follow health guidance set by governments and scientists [1], and so it is vital to identify the extent and sources of mistrust between the public and the scientific community and public health bodies [2].

Public trust in science has demonstrably declined since the beginning of the pandemic [3,4]. While trust in scientists rose early in the pandemic [5,6], trust is now lower than it was before the pandemic and shows significant political polarization [3,7–9]. Public trust in large companies, as well as in other people, declined during the pandemic in a survey conducted in the US and Netherlands [10]. Distrust manifested even in cases of games between peers, suggesting a generalized distrust [11]. Increasing numbers of people doubt official government narratives [12] and are unlikely to cooperate with government guidelines, presenting a challenge for public health measures [2].

This narrative review begins with establishing the loss of trust between the public and the scientific and medical establishments through relevant literature. Other literature is chosen on the basis of establishing deviations from scientific, bioethical, and social/democratic norms during the pandemic period which did not serve the public's interest. Our hypothesis is that these deviations were noticed by members of the public, which led to an erosion of their institutional trust.

Significant mistrust has accumulated, which is exemplified by decreased demand for childhood vaccination [13], which differs from drops in vaccination due to the inaccessibility resulting from lockdowns [14,15]. Support for vaccination rose early in the pandemic [16], but is now lower than pre-pandemic levels, and the impact cannot be entirely attributable

to closures [17,18]. Parents' intent to vaccinate has markedly decreased [19]. Vaccine hesitancy is associated with conspiratorial thinking [20–22] and distrust with conventional medicine [23]. With the increased prominence of these views, public health guidance is less likely to be followed [12].

Another proxy measure for distrust is the extent of use of complementary and alternative (CAM) medicines not offered in standard medical care. The pandemic saw a rise in such modalities [24–31], including Ayurveda [32] and herbalism [33–35], with mixed evidence on the increased use of dietary supplements [36]. The increase in CAM can be seen as a tacit desire to find an alternative to 'mainstream' medical practice [37].

Increasingly, CAM and conventional medicine are at odds; in one study, 40% of CAM practitioners surveyed in Norway said that they would not refer COVID-19 patients to a physician [38]. Antagonism has grown on both sides: government regulations increasingly target CAM practitioners. For example, the New Zealand Labour Party is introducing a 'Therapeutic Products Bill', which enables the government to regulate the sale of commonly used vitamin and mineral supplements, nutraceuticals, and natural medicines [39]. In the pandemic period, controversy existed over the safety and effectiveness of repurposed medications, nutraceuticals, supplements, and alternative therapies. Two of the most salient examples were hydroxychloroquine [40] and ivermectin [41]. Some therapeutics with strong supporting evidence, such as ivermectin for the prevention and treatment of COVID-19, were restricted under legal penalty [42], despite evidence of their treatment and prophylactic efficacy [43].

In spite of the economic pressures of the pandemic on consumers, a Romanian study demonstrated increased consumption of organic food amongst those with some organic food consumption already, but not a significant adoption of organic food consumption from those indifferent prior to the pandemic [44]. Other studies have seen an increased consumption of organic food [45–48], and entertaining heterodox beliefs on COVID-19 was a significant predictor of support for organic food [49]. Distrust of vaccines, distrust of GMOs, distrust of nuclear energy, and several other related beliefs form an associated constellation of beliefs [50].

Vaccine refusal was very common in Africa, with wilfully unvaccinated survey participants citing concerns over vaccine side effects and lack of trust in pharmaceutical industries as their major motivating factors [51]. Lack of trust features highly among people's stated reasons for not intending to get vaccinated [12,21,52–64]. Furthermore, marginalized groups with significant historical reasons for mistrust of the medical establishment and the government show lower rates of vaccination, including African Americans [65,66], Indigenous people [67–71], and Hispanics [17,19,53]. Trust on issues surrounding COVID-19 is highly partisan [72].

It has been found that people exposed to non-mainstream sources of information on COVID vaccines were less likely to get vaccinated [73]. Mainstream media strategies emphasized a single unified and authoritative message [74] and negative [75], fear- and guilt-based messaging [76–79] rather than messages likely to foster trust.

*Objectives*

We propose the research question: "How did the public health response impact public trust in scientific and public health institutions?" Having identified literature that indicates a loss of trust in institutions, we now provide evidence of departure from scientific, bioethical, and social/democratic norms during the pandemic period, which did not serve the public's interest, as a possible explanation for the decline in trust.

## 2. Reasons for Distrust

### 2.1. Censorship

In his classic work on the ethos of science, Robert K. Merton outlined four norms essential to the scientific enterprise: universalism, communism, disinterestedness, and organized scepticism [80]. At least two of these norms, universalism and organized scepticism,

are abrogated in a scientific environment where censorship and suppression of scientific findings and opposing views are rampant.

Censorship was also rife regarding topics related to SARS-CoV-2 [81]. To take the example of the COVID-19 outbreak, several dissenting scientists were accused of spreading 'misinformation' [82]. While many of their arguments were sound and came from credentialed experts [83], they were still marginalized. This was exemplified in the coordinated attempt by National Institutes for Health (NIH) director Francis Collins and National Institute for Allergies and Infectious Diseases (NIAID) director Anthony Fauci to publish a "take down" of the Great Barrington Declaration [84], a document co-authored by Drs. Martin Kulldorf, Sunetra Gupta, and Jay Bhattacharya, all highly credentialed academic experts in epidemiology [83]. Furthermore, during the pandemic, content highlighting concerns with vaccines was censored on large technology platforms including Facebook, YouTube, and Twitter [85].

In fact, many people advancing critical views towards lockdowns and mass vaccination experienced censorship, not only on social media, but from scientific journals themselves [86]. In one such episode, a manuscript for publication in the journal Current Problems in Cardiology by Jessica Rose, PhD, and Peter A. McCullough, MD, was withdrawn after publication without explanation [87]. Several examples exist of articles retracted for ostensibly political, as opposed to scientific, reasons [88,89]. While the extent of censorship has escaped the attention of much of the lay public, it has been acknowledged by human rights organizations, including Amnesty International [90].

Simply put, there is a large gap between what science is and what it is presented as. Governments pursued authoritarian policies of vaccine mandates, lockdowns, and masking for entire populations, with few exceptions. One notable exception was the country of Sweden, which did not pursue nationwide lockdowns [91]. Despite having a higher COVID-19 fatality rate early on in the pandemic than other comparable European nations, excess death was lower in Sweden than in many European countries [92–95]. Additionally, lockdown stringency, as measured by the Oxford COVID-19 Government Response Tracker [96] (as measured on 15 September 2021) was slightly, but positively associated with 2022 excess deaths (Figure 1, Table 1). This result is not statistically significant ($p = 0.177$, Table S1).

During this period, the media largely obviated their duty to question government policy, with a few exceptions emerging from alternative media. Mainstream media exposure was positively associated with disease concern [97–99]. COVID-19 regulations were advanced largely without public consultation and without disclosure of relevant conflicts of interest [100]. Education level had little impact on holding non-mainstream beliefs according to several surveys [57], and in some studies, a higher level of education was associated with vaccine hesitancy [101,102]. People with dissenting views relied on data-centric arguments to support their views, contrary to charges of social contagion of misinformation [103].

Scientists holding dissenting views often faced either explicit censorship or, more insidiously, concerns for their career which often inhibited them from questioning official narratives [86]. One example of this was lockdown policy, which had debatable benefit for limiting the spread of SARS-CoV-2 [104,105] and had significant detrimental impacts on the economy [106–109], mental health [110], education [111–113], and rates of domestic abuse [114–116]. Those taking a stand against lockdowns for any of the reasons above were often dismissed and associated with marginal viewpoints, such as speculations that 5G towers were spreading COVID-19 [117].

**Table 1.** Data for Figures 1 and 2. Average excess deaths in 2022, as a percentage of total deaths, taken from Eurostat (https://ec.europa.eu/eurostat/databrowser/view/DEMO_MEXRT__custom_309801/bookmark/table?lang=en&bookmarkId=26981184-4241-4855-b18e-8647fc8c0dd2 (accessed on 26 April 2023)). Total vaccinations per hundred people and the Oxford COVID-19 Government Response Tracker stringency index [96] (values taken on 15 September 2021), a metric of lockdown strictness, obtained from OurWorldInData.org (accessed on 26 April 2023).

| Country | Average Excess Deaths in 2022 (Percentage of Total Deaths) | Total Vaccinations per Hundred People (on 15 September 2021) | Stringency Index (on 15 September 2021) |
| --- | --- | --- | --- |
| Romania | 3.4 | 50.34 | 47.01 |
| Sweden | 4.1 | 128.65 | 33.78 |
| Hungary | 5.2 | 121.51 | 29.91 |
| Latvia | 6.5 | 83.64 | 37.96 |
| Belgium | 6.9 | 142.02 | 43.06 |
| Lithuania | 7.4 | 113.73 | 36.61 |
| Luxembourg | 7.6 | 119.83 | 37.96 |
| Czechia | 8.2 | 111.69 | 40.4 |
| Croatia | 8.8 | 83.09 | 31.81 |
| Bulgaria | 9.2 | 35.94 | 46.3 |
| Italy | 10.3 | 138.64 | 57.33 |
| Denmark | 10.4 | 145.72 | 20.37 |
| Poland | 11.1 | 92.43 | 35.77 |
| Slovenia | 11.6 | 92.3 | 43.66 |
| Norway | 12.3 | 139.59 | 38.89 |
| Spain | 12.3 | 144.99 | 42.13 |
| Netherlands | 12.5 | 127.52 | 41.67 |
| Portugal | 12.5 | 156.01 | 49.86 |
| Estonia | 13.1 | 99.69 | 27.29 |
| Slovakia | 13.2 | 83.78 | 32.65 |
| Switzerland | 13.3 | 115.12 | 43.81 |
| Germany | 14.2 | 128.23 | 36.18 |
| Austria | 15.5 | 122.16 | 46.76 |
| Greece | 15.6 | 113.56 | 73.92 |
| Finland | 16 | 133.36 | 33.8 |
| France | 16.5 | 137.18 | 43.35 |
| Malta | 17.9 | 151.69 | 43.52 |
| Iceland | 19 | 145.56 | 32.41 |
| Cyprus | 26.4 | 125.89 | 52.56 |

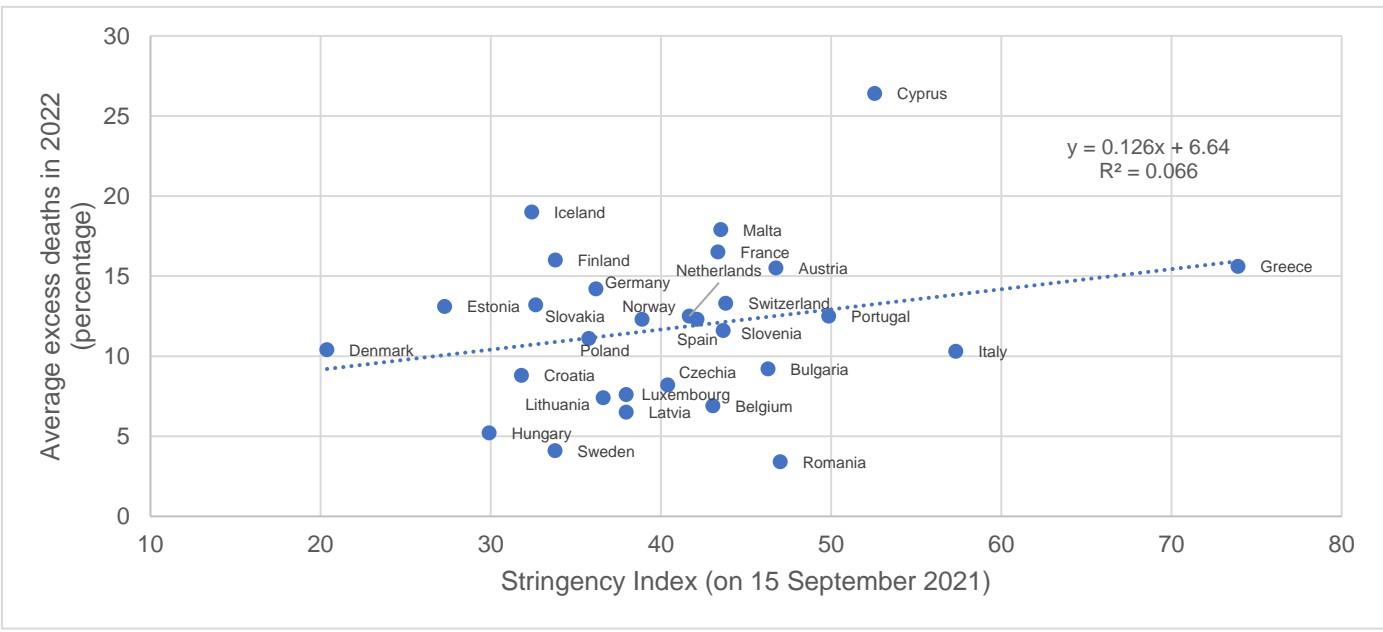

**Figure 1.** Comparison of excess mortality in European countries from January 2020 up to December 2021. Data source for excess mortality is Eurostat (https://ec.europa.eu/eurostat/databrowser/view/D EMO_MEXRT__custom_309801/bookmark/table?lang=en&bookmarkId=26981184-4241-4855-b18e-8 647fc8c0dd2, (accessed on 26 April 2023)). Data source for stringency index is OurWorldInData.org (value taken on 15 September 2021), using the stringency index developed by the Oxford COVID-19 Government Response Tracker [96]. (https://ourworldindata.org/COVID-vaccinations (accessed on 26 April 2023)).

## 2.2. Narrowness and Inflexibility of Public Health Response

The lived experience of individuals contrasts with what they were told by experts about vaccines preventing infection and transmission [118,119]. Evidence suggests that vaccination makes recipients more prone to serial reinfection, as the protection conferred by natural immunity lasts for significantly longer [120]. Highly vaccinated communities and regions still experienced COVID-19 outbreaks [121–123], and 2022 monthly excess mortality in Europe was slightly, but positively, associated with vaccination rates (Figure 2, Table 1) [124]. This result is statistically significant ($p = 0.045$, Table S1).

While symptomatic infection was the original endpoint of the clinical trial used in the Emergency Use Authorization (EUA) of the COVID vaccines [125,126], it was later stated that the vaccines were primarily intended to reduce hospitalization and death [127]. Since young people carry a much lower risk from hospitalization or death due to COVID, almost 10,000 times less fatality risk for those under 20 years old compared to those over 90 according to one Ontario study [128], it does not make sense to expose them to the risk of adverse events from these products [129].

Procrustean policies such as universal mandates that do not take into account one's individual risk, including age [130], prior infection [131], and pre-existing conditions or a lack thereof [132,133], reduced trust between the general public and the biomedical community. While booster requirements were enforced in American universities [134], research emerged showing that, based on conservative estimates of the number needed to vaccinate (NNTV) to prevent a single hospitalization from COVID-19, at least eighteen serious vaccine adverse events would occur [129]. These mandates contradict the approach that other nations, mostly European, have taken in restricting and discontinuing the use of Moderna for younger people [135,136]. Denmark even discontinued vaccinating individuals under 50 years old [137], and Switzerland recently withdrew its recommendation for continued COVID vaccination for all age groups [138].

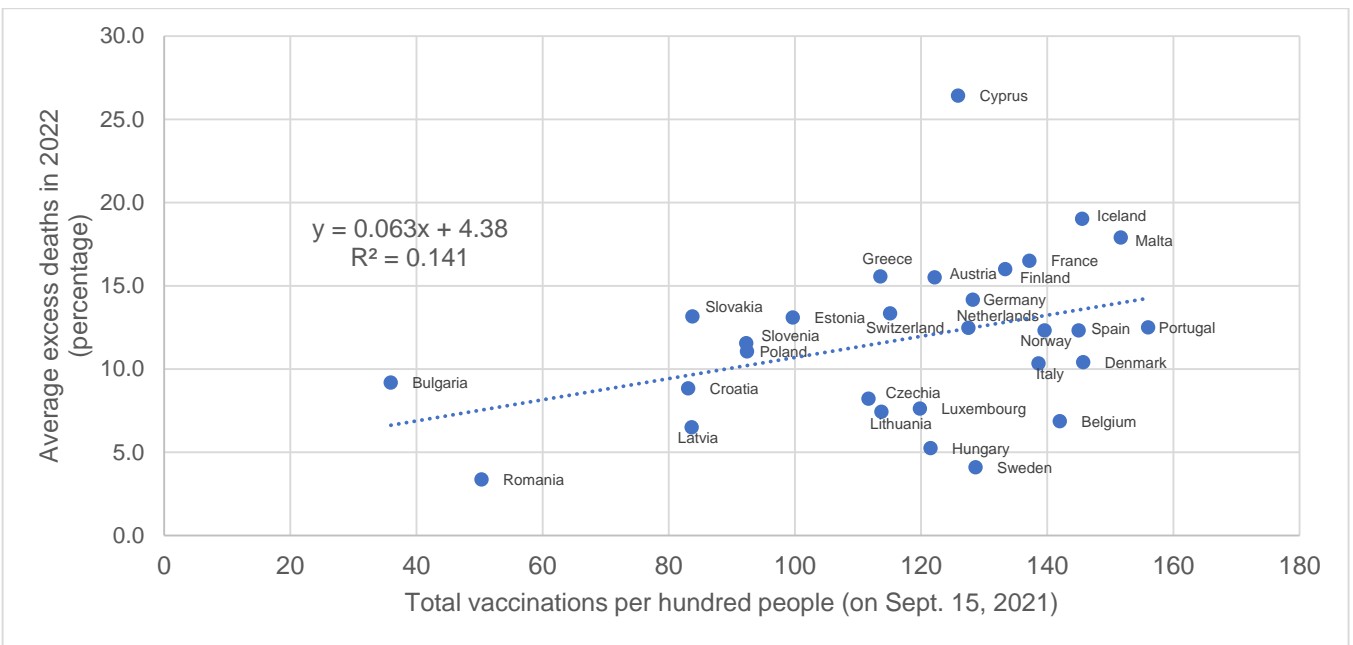

**Figure 2.** Average 2022 excess mortality in European countries vs. total vaccinations per hundred people (as of 15 September 2021). Data for excess mortality taken from Euro-Stat (https://ec.europa.eu/eurostat/databrowser/view/DEMO_MEXRT__custom_309801/bookm ark/table?lang=en&bookmarkId=26981184-4241-4855-b18e-8647fc8c0dd2 (accessed on 26 April 2023)). Monthly values from January 2022 to December 2022 are averaged to produce the average excess mortality, expressed as a percentage. Total vaccinations per hundred people on the date of 15 September 2021 are taken from OurWorldInData.org (accessed on 26 April 2023). If a value is not available for 15 September, the nearest date with data is used. See Table 1 for values.

### 2.3. Conflicts of Interest and Regulatory Capture

Loss of public trust has to be understood against the backdrop of known criminal malfeasance by the pharmaceutical industry [139,140], including the largest criminal fine in U.S. history given to Pfizer (USD 2.3 bn) [141], which was later surpassed by a USD 3 bn fine given to GlaxoSmithKline [142]. There is definite evidence of prior malfeasance, even at the level of academic research, as financial influences and conflicts of interest are known to impact research [143,144]. For example, one widely circulated one-paragraph letter in the *New England Journal of Medicine* in 1980 claimed a very low rate of opioid addiction [145,146] and was used to justify the over prescription of opioid medications, despite the letter providing no evidence. The main author, Dr. Porter, had received millions of dollars of funding from pharmaceutical companies [147]. Despite being contradicted by a wide body of evidence [148], this letter held significant sway on the field and obscured the link between opiates and addiction [146].

Conflicts of interest abound in pharmaceutical research [149,150], and pharmaceutical profits not only fund scientific journals [151,152], but also medical schools [153], patient advocacy groups [154,155], and even regulatory bodies (almost half of US FDA's annual budget [156]). Trials are also increasingly funded by the pharmaceutical companies who stand to profit from the very products under evaluation [157], resulting in significant conflicts of interest. Even intermediaries such as contract research organizations are prone to corruption [158].

### 2.4. Bioethical Violations

Another driver of people's growing distrust of science has been the experience of adverse events following administration of COVID-19 vaccines. The adverse event rate

is significantly higher than any previous vaccines [159], including those previously withdrawn due to safety concerns [160].

For example, during the swine flu epidemic of 1976–1977 in the U.S., less than 500 reported cases of Guillain–Barré syndrome (GBS) were sufficient to halt the vaccination program out of 40 million vaccines administered [161]. The prevalence of GBS following swine flu vaccination (5–10 cases per million doses) [162] is comparable to the GBS rate for COVID-19 vaccines, where estimates range from 1.8 to 53.2 cases/million doses [163]. This is just one of the many safety signals associated with COVID-19 vaccines [164–170].

Safety concerns being dismissed by public health agencies without sufficient evidentiary basis to rule out these dangers violates the bedrock bioethical principle of informed consent [171]. Considering the harms that have come to light as a result of post-marketing surveillance, the data at the time of approval was obviously insufficient to show safety, meaning that all recipients of the COVID-19 vaccines were, by definition, unable to give fully informed consent. Furthermore, mandates forced many reluctant people to receive vaccination. For example, according to one estimate, approximately $\frac{1}{4}$ of all recipients in France were otherwise unwilling but chose to receive the vaccine due to the mandate [172]. While cases where people were physically forced to be vaccinated were rare, unvaccinated individuals were often unable to work [172] or participate fully in public life [173], and fines for the unvaccinated were considered or implemented by several governments [174,175]. A core principle of informed consent is the absence of external coercion on the subject. Vaccine mandates violate the Nuremberg Code, which states [176]: "The voluntary consent of the human subject is absolutely essential. This means that the person involved should have legal capacity to give consent; should be so situated as to be able to exercise free power of choice, without the intervention of any element of force, fraud, deceit, duress, overreaching, or other ulterior form of constraint or coercion".

Since the 'consent' of a significant portion of the population was only obtained through coercion, imposing a mandate and injecting an individual with a still experimental substance constitutes a violation of the Nuremberg Code. Beyond the ethical violation, vaccine mandates and penalties for non-compliance were also predicted to damage public trust [172,177], which was borne out by subsequent research [178–180].

### 2.5. The Price of Distrust

Loss of trust is frequently blamed on people spreading contrary views or conspiracy theories rather than on the actions of the scientific and regulatory establishment itself [181]. Factors that correlate with trust [182–184] include ethnicity [185,186] (particularly for groups with a history of medical experimentation by authorities, such as African Americans [187,188]), sex [183], education, income, perceived risk [189], and cognitive disposition [190].

Distrust creates an antagonistic relationship between scientists and society and hampers cooperation. Science also becomes ineffectual in this situation, as attempts to make science-based reforms are met with hostility, and there is less support for public funding of science [191,192]. Distrust sows further distrust, as groups stop listening to each other and retreat into their respective silos. Put simply, once trust has been broken, it is difficult to restore.

Furthermore, messaging is unable to effectively 'land' for a public audience unless several communicator criteria are met, including expertise in the subject matter [193] and trust of the audience towards the communicator [194]. Understanding this necessitates movement away from the 'information-deficit' model of science communication, which is the dominant paradigm in science communication today [195], towards a different approach. The information-deficit model has proven to be an ineffective strategy even in cases when one has accurate information [196].

Taken together, there have been significant breaks from normative and proper pedagogy and science communication since the declaration of the pandemic. These breaches

have likely contributed to a less trusting relationship between the public and institutional authorities.

## 3. Discussion

During the pandemic, there was a significant drop in trust among the public in public health officials, government, media, and science. This drop is significant and will likely impact the relationship between policymakers and the public in the future, creating an antagonistic relationship and limiting future cooperation.

Trust was negatively impacted by a narrow and inflexible public health response that generated many negative externalities coupled with a high level of projected certainty and projected competence. Lockdowns caused significant collateral damage and did not achieve their stated objective of reducing mortality (Figure 1). The number of vaccinations given was positively associated with excess mortality (Figure 2, Table 1), in sharp contrast to its stated objective. We propose that many members of the public are aware of the gap between public pronouncements vaunting success of pandemic policies and their ineffectual and frequently deleterious impact.

## 4. Conclusions

Rebuilding trust necessitates accountability for offenses and lies (both of commission and omission), as well as rectification of wrongs. It means acknowledging the limitations of results, communicating that in most studies, the measured value is a proxy of the actual metric of interest (such as mice antibody levels and no human trials used in the approval of bivalent boosters by the FDA [197]), reporting uncertainties and the possibility that the response may change with new information [198,199]. This also guards against holding onto models too tightly when they need to be updated.

It also means open communication and free speech as fundamental principles. Conflicts of interest must be disclosed and investigated where relevant, and firings, as well as legal and criminal accountability, must be enforced when violations are present to maintain scientific and medical integrity.

Additionally, transparency and openness need to be operative principles of science. Where there are raw data, they should be accessible to an interested researcher (after appropriate steps are taken to preserve subject confidentiality) [200], and analytic procedures must be clearly posted to enable replicability. The FAIR guidelines (findable, accessible, interoperable, and reusable) have been developed for this purpose, and they should guide publishing in the future [201]. Not only is there an added benefit for a field adopting open data policies [202], but there is also greater trust engendered by the openness [203].

We do not know to what extent such an approach would be effective in restoring trust, and it may not disabuse all members of the public of their distrust, but current efforts have proven ineffectual. Humanity faces multiple converging crises in health, ecology, and in the wider social fabric. A continued relationship of antagonism between policymakers and large swathes of the public hampers our ability to face the challenges ahead.

**Supplementary Materials:** The following supporting information can be downloaded at: https://www.mdpi.com/article/10.3390/biomed3020023/s1, Table S1: Statistical parameters for the fits in Figures 1 and 2, calculated using the LINEST function in Microsoft Excel.

**Author Contributions:** Conceptualization, M.T.J.H.; writing—original draft preparation, M.T.J.H. and J.G.; writing—review and editing, M.T.J.H. and J.G.; project administration, M.T.J.H. All authors have read and agreed to the published version of the manuscript.

**Funding:** This research received no external funding.

**Institutional Review Board Statement:** Not applicable.

**Informed Consent Statement:** Not applicable.

**Data Availability Statement:** Publicly available datasets were analyzed in this study. This data can be found here: (https://ec.europa.eu/eurostat/databrowser/view/DEMO_MEXRT__custom_3098 01/bookmark/table?lang=en&bookmarkId=26981184-4241-4855-b18e-8647fc8c0dd2 (accessed on 1 April 2023)) and (OurWorldInData.org (accessed on 1 April 2023)).

**Acknowledgments:** The authors would like to thank David Charalambous for discussions related to this manuscript.

**Conflicts of Interest:** The authors declare that the research was conducted in the absence of any commercial or financial relationships that could be construed as a potential conflict of interest.

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
