# Peer review of "Public Health Needs the Public Trust: A Pandemic Retrospective"

_2673-8430, doi:10.3390/biomed3020023_

Round 1

Reviewer 1 Report (Previous Reviewer 2)

Thanks for the author's efforts to revise. After careful reading, check the standard of the journal. I have no other opinion. Agree to publish.

Thanks for the author's efforts to revise. After careful reading, check the standard of the journal. I have no other opinion. Agree to publish.

Author Response

We thank the reviewer for his or her comments. We have made some minor edits to the English in the report.

Reviewer 2 Report (New Reviewer)

1. No mention or description is provided on the research design, methods  and related issues. Even though this is a perspective piece, this is still relevant. 

2. Use of 'hard data' to support the assertions presented would have enhanced  the value of this paper. Authors are encouraged to include tables or diagrams with relevant databon the key points being made. 

3. The statement which refers to bioethical violations- i.e. consent being obtained only through coercion is a particularly strong and potentially misleading statement as vaccination programmes required individual consent. The authors should consider amending this assertion by indicating that with media penetration on the subject, it is likely that the global and regional environments may have influenced persons to consent to accepting vaccination. 

4. It would be very useful for the authors to highlight contrasting evidence/outcomes on lockdowns and mass vaccination  and views on the subject - i.e.  instances where the literature provides epidemiological data on rates of infection and hospitalization comparing regions with  high vs. low rates of compliance on protocols; socio-cultural and economic factors, etc. which contributed to  differences in outcomes;

5. Authors are encouraged to include a discussion on area/regional/country differences in views on lock-downs and mass vaccination (supportive vs. non-supportive)and related health outcomes which would enhance the objectivity and ethical 'soundness' of this article. 

6. Use of acronyms - e.g. NIH and NIAID should be spelled out at first use 

Author Response

  1. No mention or description is provided on the research design, methods  and related issues. Even though this is a perspective piece, this is still relevant. 

It has been added that the article is a narrative review.

  1. Use of 'hard data' to support the assertions presented would have enhanced  the value

 of this paper. Authors are encouraged to include tables or diagrams with relevant databon the key points being made. 

A figure demonstrating comparatively low excess deaths in Sweden (vs the rest of Europe) is included.

  1. The statement which refers to bioethical violations- i.e. consent being obtained only through coercion is a particularly strong and potentially misleading statement as vaccination programmes required individual consent. The authors should consider amending this assertion by indicating that with media penetration on the subject, it is likely that the global and regional environments may have influenced persons to consent to accepting vaccination. 

Added that we are not talking about forced vaccination, but the use of coercion.

“While truly forced vaccination was rare, unvaccinated individuals were often unable to work [1] [1–3], participate fully in public life [1], and fines for the unvaccinated were con-sidered or implemented by several governments [1,2].”

  1. It would be very useful for the authors to highlight contrasting evidence/outcomes on lockdowns and mass vaccination  and views on the subject - i.e.  instances where the literature provides epidemiological data on rates of infection and hospitalization comparing regions with  high vs. low rates of compliance on protocols; socio-cultural and economic factors, etc. which contributed to  differences in outcomes;

The comparison of Sweden with other countries in Europe was added for the issue of lockdowns.

On vaccinations, we include the Aarstad and Kvitastein, 2023, showing a non-negative association between monthly excess deaths in Europe in 2022 and vaccination coverage.

  1. Authors are encouraged to include a discussion on area/regional/country differences in views on lock-downs and mass vaccination (supportive vs. non-supportive)and related health outcomes which would enhance the objectivity and ethical 'soundness' of this article. 

A comparison between the excess mortality of Sweden, which did not use lockdowns, and the rest of Europe is shown, where Sweden had one of the lowest 2022 rates of excess mortality in Europe. Additionally, a reference to Aarsted and Kvitastein, 2023, is included, which shows a slight positive correlation between vaccination uptake and monthly excess mortality in 2022 in Europe.

Aarstad, J.; Kvitastein, O.A. Is There a Link between the 2021 COVID-19 Vaccination Uptake in Europe and 2022 Excess All-Cause Mortality? 25-31 2023, doi:10.21276/apjhs.2023.10.1.6.

  1. Use of acronyms - e.g. NIH and NIAID should be spelled out at first use 

This has been corrected.

We thank the reviewers for their comments.

Round 2

Reviewer 2 Report (New Reviewer)

Results & Discussion:

While the authors provided a figure indicating lower excess deaths in Sweden, it is noteworthy that other countries on the lower end included Norway, Cyprus and Finland, even Iceland with a negative rate.  It is instructive to provide commentary on the status of these other countries. Are the same assumptions to be made of their lockdown policies, etc as of Sweden. What factors accounted for their lowered rates of mortality? How does this data/figure substantially distinguish Sweden from other country experiences?

Furthermore, for those countries with higher mortality rates some commentary/review of their policies should be done. A table is recommended to include data on policies for each of the countries, alongside the morality rates (excess). 

This is necessary for the authors to present objective data to support their hypothesis that Covid 19 public health measures are associated with mortality rates. 

Methodology: Authors have indicated this is a narrative review. However a specific methodology section detailing the methodology used to select articles is relevant. Authors may consider the SANRA approach - Scale for Assessment of Narrative Review Articles, or some such relevant approach/methodology.

Authors have reviewed the literature around safety, distrust and discuss bioethical principles of good science, such as transparency, communication and public trust did not exclusively provide a  conclusion to the research question as it relates specifically to the pandemic of Covid 19.  

Author Response

We thank the reviewers for their valuable and constructive comments. 

While the authors provided a figure indicating lower excess deaths in Sweden, it is noteworthy that other countries on the lower end included Norway, Cyprus and Finland, even Iceland with a negative rate.  It is instructive to provide commentary on the status of these other countries. Are the same assumptions to be made of their lockdown policies, etc as of Sweden. What factors accounted for their lowered rates of mortality? How does this data/figure substantially distinguish Sweden from other country experiences?

We demonstrate a scatterplot of Average 2022 excess mortality as a function of stringency index and observe a slight, but positive correlation. In the graph of the previous version, we mistakenly used the entry “Estimating excess mortality due to the COVID-19 pandemic (Jan 2020 up to Dec 2021)” from the UK Office of National Statistics (ONS) https://www.ons.gov.uk/peoplepopulationandcommunity/birthsdeathsandmarriages/deaths/articles/comparingdifferentinternationalmeasuresofexcessmortality/2022-12-20. I.e. these were excess deaths due to COVID-19, and not overall excess deaths. We are using the data from EuroStat for excess mortality measures in 2022 https://ec.europa.eu/eurostat/databrowser/view/DEMO_MEXRT__custom_309801/bookmark/table?lang=en&bookmarkId=26981184-4241-4855-b18e-8647fc8c0dd2. We argue that lockdowns may have had limited utility in preventing the spread of Covid-19, but any benefits to overall mortality were undone by their harms

Furthermore, for those countries with higher mortality rates some commentary/review of their policies should be done. A table is recommended to include data on policies for each of the countries, alongside the morality rates (excess). 

This is necessary for the authors to present objective data to support their hypothesis that Covid 19 public health measures are associated with mortality rates. 

We have included two figures and a table. The figures show the slight positive correlation of stringency index (taken on Sept.15, 2021) with Average 2022 excess mortality and the slight positive correlation of vaccinations per hundred people with Average 2022 excess mortality. We are not claiming a causal relationship here, we are claiming that the interventions had negative consequences outside of reducing COVID-19 mortality.

Methodology: Authors have indicated this is a narrative review. However a specific methodology section detailing the methodology used to select articles is relevant. Authors may consider the SANRA approach - Scale for Assessment of Narrative Review Articles, or some such relevant approach/methodology.

Authors have reviewed the literature around safety, distrust and discuss bioethical principles of good science, such as transparency, communication and public trust did not exclusively provide a conclusion to the research question as it relates specifically to the pandemic of Covid 19.  

We propose the research question. How did the public health response impact public trust in scientific and public health institutions? Our working hypothesis is that trust was negatively impacted by a largely ineffectual public health response coupled with a high level of projected certainty and projected competence. We propose that the public is aware of the ruse due to the visible failures of pandemic policy. Some examples of these failures were: unsafe and ineffective vaccines, economically destructive lockdown policies which were ineffective at preventing deaths, unscientific mask mandates. Simultaneously alongside the apparent incompetence of health authorities, a strong unified message was presented that communicated zero or very minimal levels of uncertainty when the certainty should have been very low, or directed towards the opposite hypothesis.

Our methodology involves identifying literature which is first indicative of the loss of trust and later provides evidence of departure from scientific, bioethical and social/ democratic norms during the pandemic period, which did not serve the public’s interest.

We thank Reviewer #2 for his or her comments.

Round 3

Reviewer 2 Report (New Reviewer)

The following are the most recent observations and comments on the revised paper.

1. Methodology:

In their response to the query on methodology, authors stated the following. However this is not indicated in the revised paper. A more comprehensive report on the narrative methods used is required

"Our methodology involves identifying literature which is first indicative of the loss of trust and later provides evidence of departure from scientific, bioethical and social/ democratic norms during the pandemic period, which did not serve the public’s interest". 

2. Hypothesis/Research Questions/Objectives:

Hypothesis should be stated in a subsection on study objectives, notably as stated by the authors:-

"We propose the research question. How did the public health response impact public trust in scientific and public health institutions?"

3. Conclusions:

Authors have addressed statements relevant to a conclusion :However these arguments are not conclusively stated in the paper.as in the following

"... trust was negatively impacted by a largely ineffectual public health response coupled with a high level of projected certainty and projected competence. We propose that the public is aware of the ruse due to the visible failures of pandemic policy. Some examples of these failures were: unsafe and ineffective vaccines, economically destructive lockdown policies which were ineffective at preventing deaths, unscientific mask mandates". Would the data provided in Table 1 and Figure 1 & 2 be relevant here as part of the outcomes which may have lead to a distrust of the policies? Authors should comment.

3. Tables and analysis

See previous comments from review - "Furthermore, for those countries with higher mortality rates some commentary/review of their policies should be done".

Authors have presented data in Figure 1 & 2 as well as Table 1. However the analysis of these is inadequate and should be expanded in a paragraph or two. Authors should explicitly state if any of these associations are statistically significant. In the absence of such evidence the authors should use appropriate  terms -e.g.  likely, probable, likelihood, etc.  to guard against any potential interpretation of causation.

This comment is also raised in relation to the authors' own words/response - "We are not claiming a causal relationship here, we are claiming that the interventions had negative consequences outside of reducing COVID-19 mortality".

4. Recommended reorganization of table:

In Table 1 re-arranging the data would make a more visible and compelling argument of probable association  or relationships (particularly as no statistically significant results are shared)

Author Response

This manuscript is a resubmission of an earlier submission. The following is a list of the peer review reports and author responses from that submission.

Round 1

Reviewer 1 Report

Overall I find this paper raises an important issue, and agree with their general diagnosis of the problems and it’s causes.  I would accept the paper in it’s present form but still I offer multiple suggestions that I believe, could potentially improve the paper.  I would allow the authors to decide which of the suggestions they believe would improve their paper.   

Regarding lines 48-49: Some therapeutics with strong supporting evidence, such as ivermectin for the prevention and treatment of covid-19, were restricted under legal penalty.

The strength of evidence of ivermectin efficacy is a highly debated topic, without a more indepth explanation of the strong evidence supporting ivermectin, to avoid this, this statement can be slightly changed to refer to how Therapeutics with decades of evidence of safety, were restricted under legal penalty, preventing physicians from prescribing medication off label to their patients, harming the physician patient relationship.    

Regarding line 79: “Censorship was also rife regarding all topics related to SARS-CoV-2[65]. To take the ex-78 ample of the COVID-19 outbreak, there were several non-mainstream scientists accuse.”

, as they were mainstream in the sense of being affiliated with top universities.  Potentially these scientists would be better described as scientists who criticized government public health policy, or scientists with viewpoints different than governmental agencies were censored. 

Regarding lines 86-88 “In fact, many people advancing non-mainstream views, especially those challenging the 86 need for, safety and efficacy of products known as COVID-19 vaccines, saw censorship, 87 not only on social media, but by scientific journals themselves.

Here again the term “non-mainstream” does not capture this concept perfectly.  An alternative option is “many people advancing views that questioned government policy.”

Regarding lines 101-102 “Responsibility certainly falls on the media, and governments for pursuing a singular, one-size-fits-all strategy of vaccine mandates, lockdowns, and masking for entire populations, with few exceptions.”

It would make sense to differentiate the difference in responsibility between the governments and the media, in that the Government would be responsible for pursuing the policies.  While the media failed in their responsibilities to question the Governments policies. 

Regarding lines 108 and 109” However, science and scientists are not blameless in their lack of questioning official guidance when it contradicted science. 

I think if mentioning silence from scientists it would be important to mention how the threat of having one’s articles or spoken word when labeled as misinformation or censored threatens the careers of academics, and creates an environment that encourages self-censorship.    

Regarding lines 116 & 117

In fact, vaccination makes recipients more prone to serial reinfection, as the protection conferred by natural immunity lasts for significantly longer

Upon quick review of the reference for this sentence, I was unable to easily identify evidence for this statement.  While the data supporting this statement may be contained within the reference, I believe an alternative example that better represented people’s lived experience, such as the widespread rate of infection and transmission that was occurring in the general public, best exemplified by the COVID-19 outbreak that occurred in provincetown in a highly vaccinated population. 

Regarding lines 152-154

The adverse event rate is significantly higher than any previously administered vaccine, and is even much higher than the rates of vaccines previously withdrawn due to safety concerns[123].  

I believe the authors are referring to serious adverse events, rather than adverse events in general.   Given the limited data that has been made public on vaccine induced serious adverse events, existing rates are only estimates, and it may be better to describe this point slightly differently as stating a higher number of safety signals for serious adverse events have been identified that exceeds the safety signals of previously withdrawn vaccines.  In addition studies have identified evidence suggesting the serious adverse event rate is potentially higher than any previously approved vaccine. 

Regarding lines 178-180 and 186-189

This sentence appears to be accidentally repeated. 

Reviewer 2 Report

The topic of this paper is quite meaningful. The author has read a lot of literature, which is worthy of recognition. But after I read it carefully. I personally feel that there are some structural problems in this paper. I don't think this paper has reached the level of publication.

1. The paper is not focused enough. During the COVID-19 epidemic, there were many perspectives of public trust in public health, and there were significant differences between different countries. It is suggested to focus on the public in a country or a specific region for research. At present, the research object is not specific.

2. Although this paper is a summary paper. But there is no innovation in the method, which is more traditional induction.

3. Many paragraphs in the paper are too trivial. It is suggested that similar views be summarized in one paragraph.

4. In the second part, the author summarizes the reasons for public distrust of public health. However, it is not clear why these reasons occur. Or, what is the standard for the author to analyze such reasons? If you don't explain clearly, you can find many reasons.

5. In the third part, the issues discussed are not very focused and the evidence is too few. The author's subjectivity is stronger.

Therefore, I personally believe that this paper has great problems and it is not recommended to publish it.